# Flupyradifurone Exhibits Greater Toxicity to the Asian Bumblebee *Bombus lantschouensis* Compared to the European Bumblebee *Bombus terrestris*

**DOI:** 10.3390/insects16050455

**Published:** 2025-04-25

**Authors:** Chunting Jie, Hong Zhang, Ziyu Zhou, Zhengying Miao, Bo Han, Baodi Guo, Yi Guo, Xiao Hu, Shahid Iqbal, Bingshuai Wei, Jiaxing Huang, Pingli Dai, Jiandong An

**Affiliations:** 1State Key Laboratory of Resource Insects, Key Laboratory of Insect-Pollinator Biology of Ministry of Agriculture and Rural Affairs, Institute of Apicultural Research, Chinese Academy of Agricultural Sciences, Beijing 100193, China; jct970115@163.com (C.J.); angela99517zzy@163.com (Z.Z.); 17720817366@163.com (B.H.); baodi_guo@163.com (B.G.); guoyi63671@163.com (Y.G.); xiaohu0607@yeah.net (X.H.); shahidiqbal525592@gmail.com (S.I.); w3295780874@163.com (B.W.); huangjiaxing@caas.cn (J.H.); daipingli@caas.cn (P.D.); 2Gansu Provincial Beekeeping Technology Extension Station, Tianshui 741022, China; gsyfs@163.com

**Keywords:** bumblebee, acute toxicity, chronic toxicity, risk assessment

## Abstract

Bumblebee pollination is crucial for maintaining the balance of both natural and agricultural ecosystems. However, a growing body of research indicates a significant decline in bumblebee diversity across numerous countries and regions in recent years. One of the primary drivers of this decline is the widespread use of pesticides, especially neonicotinoids. As a potential alternative to neonicotinoids, flupyradifurone has emerged as a promising agent for pest control, with initial assessments suggesting that it may be relatively safe for bees. Despite this, there is a lack of comprehensive data on the lethal and sublethal effects of flupyradifurone on bumblebees, particularly wild species. This study aims to address this gap by evaluating the toxicity and associated risks of flupyradifurone on the commercially available bumblebee *Bombus terrestris* and Asian wild species *B. lantschouensis*. Our findings show that flupyradifurone does not meet the specific protection criteria required for these two bumblebee pollinators, with *B. lantschouensis* exhibiting greater sensitivity to the pesticide than *B. terrestris*. This research contributes essential data on the lethal effects and risk assessments of flupyradifurone on *B. terrestris* and *B. lantschouensis*, laying the groundwork for further investigations into the environmental impacts of this pesticide on bumblebee populations.

## 1. Introduction

Bumblebees play a crucial role as pollinators in both natural and agroecosystems [1,2], but their diversity has significantly declined in recent years in many regions worldwide [3,4,5,6,7,8,9]. Among the various factors contributing to this decline, pesticide use is a major concern [10,11,12]. Assessing the toxicity and risk of pesticides on bumblebees is critical for their conservation.

Neonicotinoid insecticides, which target acetylcholine receptors in insects, are highly effective and broad-spectrum and are the most widely used insecticides in agriculture. They have low toxicity to mammals and good systemic absorption in plants, making them a popular choice for pest control [13]. However, numerous studies have raised concerns about their harmful effects on non-target organisms, particularly bee pollinators [14,15,16,17]. In response, many countries have implemented evaluation systems and restrictions for neonicotinoids [18,19]. This has highlighted the urgent need for alternative insecticides that are safer for pollinators.

Flupyradifurone, a novel insecticide developed by Bayer, selectively binds to nicotinic acetylcholine receptors in insects [20]. It has broad-spectrum efficacy and is suitable for the prevention and control of various pests in vegetables and fruits [21]. Although some studies suggest that flupyradifurone is relatively ‘safe’ for honey bees when used at recommended doses [22], other studies have shown that it has a range of negative effects on honey bees. For instance, exposure to a field-realistic dose of flupyradifurone over 10 days significantly increased the mortality rate of honey bee *Apis mellifera* colonies [23]. When the duration of chronic toxicity testing was extended to 30 days, honey bee larvae exposed to sublethal doses of flupyradifurone exhibited significantly reduced survival rates, and the expression of genes related to immunity and detoxification was altered [24]. Tan et al. [25] demonstrated that flupyradifurone not only impaired the olfactory learning ability of adult *A. cerana* honey bees but also affected the larvae. Furthermore, flupyradifurone negatively impacted the flight and thermoregulation abilities of honey bees, increased oxidative stress, and induced apoptosis [26,27].

In addition to honey bees, an increasing number of studies have demonstrated the threats of flupyradifurone to bumblebees. Siviter and Muth [28] found that flupyradifurone negatively affects the olfactory, visual learning, and memory abilities of *B. impatiens*. Additionally, flupyradifurone exposure can reduce the reproductive output of *B. terrestris* and *B. impatiens*, leading to prolonged colony developmental times and increased larval mortality [29,30]. Furthermore, the negative effects of flupyradifurone on bumblebees could not be alleviated by improved nutrition [31].

Although increasing attention has been given to the sub-lethal effects of flupyradifurone on bumblebees, a significant knowledge gap remains regarding its lethal effects from both acute and chronic exposure. The exposure concentrations or doses in the referenced studies were often based on acute toxicity to honey bees or pesticide residue data, and some studies even used randomly selected concentration gradients [32]. This variability in exposure doses across studies makes it difficult to directly compare findings and draw general conclusions. Therefore, it is important to determine the acute and chronic toxicity of flupyradifurone to bumblebees in order to establish a reliable exposure dose for further laboratory or field risk assessments.

Current research on the impacts of flupyradifurone on bumblebees has focused on commercially used species for pollination, such as *B. terrestris* in Europe and *B. impatiens* in North America. However, there remains a significant lack of toxicity data regarding flupyradifurone’s effects on wild bumblebee species. To address this knowledge gap, we selected the commonly used commercial bumblebee *B. terrestris* and the Asian native wild bumblebee *B. lantschouensis* (field surveys on bumblebees indicate that this species is primarily distributed in northern China [33]). The latter was chosen due to its potential for commercial pollination in China [34,35]. The objectives of this study were (1) to determine the acute and chronic oral toxicity of flupyradifurone to workers of *B. terrestris* and *B. lantschouensis* and (2) to assess the risks to these bumblebee species using two risk assessment equations—the Hazard Quotient (HQ) and the Exposure Toxicity Ratio (ETR).

## 2. Materials and Methods

### 2.1. Test Chemical

Flupyradifurone standard (99.23% pure, Dr. Ehrenstorfer Company, Augsburg, Deutschland, Germany) was dissolved in sterilized deionized water (diH_2_O) to prepare a stock solution with a concentration of 2000 μg/mL, which was then stored at −20 °C. Feeding solutions of different flupyradifurone concentrations were prepared from the stock solution using a 50% (*w*/*v*) sucrose solution. For the acute oral toxicity tests, concentrations of flupyradifurone were determined through range-finding trials using 6 colonies of each bumblebee species. The acute exposure concentrations for *B. terrestris* were 1518.75, 1012.5, 675, 450, and 300 μg/mL, and for *B. lantschouensis* were 800, 400, 200,100 and 50 μg/mL. For the chronic oral toxicity tests, the concentrations were based on the results of the acute toxicity tests. The chronic exposure concentrations for *B. terrestris* were 480, 240, 120, 60, 30, and 15 μg/mL, and for *B. lantschouensis* were 320, 160, 80, 40, 20, and 10 μg/mL. Quantitative analysis of the solutions at different concentrations was performed using GC-MS to minimize the deviation between the actual and prepared concentrations. In this study, a 50% sucrose solution was used as the control group, and dimethoate (200 μg/mL) was used as the reference substance.

### 2.2. Test Organisms

In this study, both the commercial bumblebee *B. terrestris* and the wild bumblebee *B. lantschouensis* were tested (Figure 1). The *B. terrestris* colonies were purchased from Woofuntech Biocontrol (Hengshui, Hebei, China), while the *B. lantschouensis* colonies were first-generation offspring from wild queens collected in Gansu Province, China, by the Institute of Apicultural Research, Chinese Academy of Agricultural Sciences. For each species, 25–30 normal colonies, each containing a healthy queen, approximately 60 to 70 workers, and no males, were selected for sampling. The colonies were maintained at 26 ± 2 °C, 60 ± 3% relative humidity under dark conditions. Colonies were fed every other day with a sucrose solution (50%, *w*/*v*) and a pollen paste (a mixture of fresh pollen from honey bee hives and pure water).

Healthy, active adult worker bumblebees were randomly selected from the colonies and weighed individually. Bumblebees of similar body size were placed individually in isolation cages (Figure 2A). The cages containing the bumblebee were positioned next to each other at a distance of approximately 2–3 cm to allow for olfactory and visual contact between individuals (Figure 2B). These isolated bumblebees were provided with a 50% (*w*/*v*) sucrose solution and placed in an incubator for at least 24 h to acclimate to the test conditions (Figure 2C, incubator temperature: 26 °C, relative humidity: 60%).

### 2.3. Oral Toxicity Test

The protocol for the oral acute and chronic toxicity tests for bumblebees was adapted from the Organization for Economic Co-Operation and Development [36,37]. Workers in the isolation cages were starved for 15 h, then randomly divided into seven treatment groups for the acute toxicity test (five concentrations of flupyradifurone, one control group, and one reference substance group) and eight treatment groups for the chronic toxicity test (six concentrations of flupyradifurone, one control group, and one reference substance group), with 30 workers in each group. Bumblebees in each group were fed the corresponding test chemical using a pipette to ensure each bee consumed 20 μL of solution (Figure 2D). In the acute toxicity test, after being fed the test chemical, bumblebees were returned to the incubator and provided with a 50% sucrose solution. Mortality in the different treatment groups was observed and recorded at 4, 24, 48, and 72 h after consumption of the chemical solution, with only the 72-h median lethal concentration (LC_50_) and 72-h median lethal dose (LD_50_) values reported in this study. In the chronic toxicity test, bumblebees were fed with 20 μL of the corresponding test chemical solution once a day for 10 consecutive days. After consuming the treated sucrose solution, bumblebees were returned to the incubator and provided with 50% sucrose solution. Mortality was recorded daily over the 10-day period, with the 10-day median lethal dietary dose (LDD_50_) and no observed adverse effect concentration (NOAEC) value reported in this study.

### 2.4. Risk Assessment

The Hazard Quotient (HQ) proposed by the European and Mediterranean Plant Protection Organization (EPPO) [38] and the Exposure Toxicity Ratio (ETR) published by the European Food Safety Authority (EFSA) [39] were used to assess the risks of oral flupyradifurone exposure to bumblebees *B. terrestris* and *B. lantschouensis*. The HQ value represents the ratio of the amount of active ingredient (A.I.) in a commercial product applied to the crop (g A.I./ha) to the acute oral LD_50_ of the active ingredient (Equation (1)).(1)$$\mathrm{Hazard}\;\mathrm{Quotient}\;\left(\mathrm{HQ}\right)=\frac{\mathrm{Application}\;\mathrm{Rate}\;(\frac{g\;A.I.}{\mathrm{ha}})}{\mathrm{Acute}\;\mathrm{Oral}\;{\mathrm{LD}}_{50}\;(\frac{\mu g\;A.I.}{\mathrm{bee}})}$$

In this equation, the application rate was obtained from the maximum recommended field use rate of flupyradifurone within commercial products registered by the Ministry of Agriculture and Rural Affairs of the People’s Republic of China (102 g/ha). The 72-h LD_50_ value from the acute oral toxicity test was used in this calculation. If HQ > 2500, it indicates a high risk posed by the pesticide to bumblebees; if 50 < HQ ≤ 2500, it indicates a medium risk; if HQ ≤ 50, it indicates a low risk.

The ETR equation is used in both the screening step and first-tier assessments. The equation varies depending on the application type (spray application, seed application, or granular application), with only the spray application of flupyradifurone currently approved in China. Therefore, the ETR equation for spray application was used in this study. In the screening step assessment, the ETR is calculated using the application rate, a shortcut value (SV), and the oral LD_50_ (Equation (2)).(2)$$\mathrm{Exposure}\;\mathrm{Toxicity}\;\mathrm{Ratio}\;\left(\mathrm{ETR}\right)=\frac{\mathrm{Application}\;\mathrm{Rate}\;\left(\frac{kg\;A.I.}{\mathrm{ha}}\right)\times \mathrm{Shortcut}\;\mathrm{Value}\;(\mathrm{SV})}{\mathrm{Oral}\;{\mathrm{LD}}_{50}\;(\frac{\mu g\;A.I.}{\mathrm{bee}})}$$

In this equation, the application rate was obtained from the maximum recommended field use rate of flupyradifurone in commercial products registered by the Ministry of Agriculture and Rural Affairs of the People’s Republic of China (0.102 kg/ha). This evaluation method accounts for the application method, where the SV value differs for two spraying situations (downward spraying and sideward spraying, Appendix A). The 72-h LD_50_ value from the acute oral toxicity test was used in the acute oral exposure assessment, while the 10-day LDD_50_ value obtained from the chronic oral toxicity test was used in the chronic oral exposure assessment (Appendix A).

If the ETR value is less than or equal to the trigger value (TV), it indicates that the pesticide poses minimal threat to bumblebees and meets the specific protection requirements. If the ETR exceeds the TV, a first-tier assessment is required. In the first-tier assessment, the SV values were refined based on different risk scenarios (e.g., risk from foraging on the treated crop, adjacent crops, or weeds in the treated field, as well as foraging in the field margin, in permanent crops the following year, or on succeeding crops for annual crops). In this study, only the risk from foraging on the treated crop was considered. The SV value also varies between acute and chronic exposure, and a new variable, the Exposure factor (Ef), was introduced into the equation (Equation (3), Appendix A).(3)$$\mathrm{Exposure}\;\mathrm{Toxicity}\;\mathrm{Ratio}\left(\mathrm{ETR}\right)=\frac{\mathrm{Application}\;\mathrm{Rate}\;\left(\frac{\mathrm{kg}\;A.I.}{\mathrm{ha}}\right)\times \mathrm{Exposure}\;\mathrm{Factor}\;\left(\mathrm{Ef}\right)\times \mathrm{Short}\;\mathrm{Value}\;\left(\mathrm{SV}\right)\times \mathrm{twa}}{\mathrm{Oral}\;{\mathrm{LD}}_{50}\;(\frac{\mu g\;A.I.}{\mathrm{bee}})}$$

### 2.5. Statistical Analysis

Mortality rates were adjusted for solvent control effects using Abbott’s correction formula. LC_50_ values were derived from log concentration/response curves through probit analysis. The 95% confidence intervals of LC_50_ values were calculated by least-squares regression analysis, with relative growth rates (expressed as percentage of control) regressed against the logarithm of compound concentrations. Similarly, LD_50_ and LDD_50_ values were determined from log dose/response curves using probit analysis [40]. The survival data were analyzed by Kaplan–Meier survival analysis, and log-rank tests were used to compare the survival rate of bumblebee workers from flupyradifurone-exposed groups and control group, where the highest concentration showing no significant difference compared to the control group was the NOAEC. Acute toxicity data were analyzed using the Data Analysis package in Microsoft Excel 2019 and chronic toxicity data were analyzed in SAS 9.2 software.

## 3. Results

### 3.1. Acute Oral Toxicity of Flupyradifurone to the Two Bumblebee Species

The results showed that the acute oral toxicity of flupyradifurone to *B. lantschouensis* was higher than that to *B. terrestris*. The 72-h LC_50_ values of flupyradifurone for workers of *B. terrestris* and *B. lantschouensis* were 1400 μg/mL and 250 μg/mL, respectively. The 72-h LD_50_ values for workers of *B. terrestris* and *B. lantschouensis* were 28 μg/bee and 5.1 μg/bee, respectively (Table 1).

### 3.2. Chronic Oral Toxicity of Flupyradifurone to the Two Bumblebee Species

The survival rates of bumblebees fed different concentrations of flupyradifurone sucrose solution varied significantly (Log-rank test: *B. terrestris*: χ^2^ = 240.121, df = 7, *p* < 0.0001; *B. lantschouensis*: χ^2^ = 266.379, df = 7, *p* < 0.0001). The results indicated that the chronic oral toxicity of flupyradifurone to *B. lantschouensis* was higher than that to *B. terrestris* (Table 2, Figure 3). The 10-day LDD_50_ values for workers of *B. terrestris* and *B. lantschouensis* were 3.3 μg/bee/day and 0.7 μg/bee/day, respectively.

The NOAEC (No Observed Adverse Effect Concentration) of flupyradifurone for *B. terrestris* workers was 60 μg/mL, while for *B. lantschouensis*, it was 20 μg/mL (Figure 3; Appendix A). For *B. terrestris*, no significant differences were observed in the average survival rates between workers fed lower concentrations of flupyradifurone sucrose solutions (15 μg/mL, 30 μg/mL, 60 μg/mL) and the control group (Figure 3A; Appendix A), with survival remaining at 85% after 10 days. The survival rates of workers fed flupyradifurone sucrose solutions with concentrations of 120 μg/mL, 240 μg/mL, and 480 μg/mL on the 10th day were significantly lower than those of the control group (Figure 3A; Appendix A). For *B. lantschouensis*, the average survival rates of workers fed lower concentrations of flupyradifurone sucrose solution (10 μg/mL, 20 μg/mL) and pure sucrose solution (control group) on the 10th day were higher than 85%, with no significant differences between the lower concentrations and the control group (Figure 3B, Appendix A). The survival rates of workers fed flupyradifurone sucrose solution with concentrations of 40 μg/mL, 80 μg/mL, 160 μg/mL, and 320 μg/mL on the 10th day were significantly lower than those of the control group (Figure 3B, Appendix A).

### 3.3. Risk Assessment of Flupyradifurone to the Two Bumblebee Species

The equations of HQ and ETR produced different distributions of value across risk classifications. The HQ values for the acute oral toxicity of flupyradifurone to *B. terrestris* and *B. lantschouensis* were both lower than 50, indicating a low risk (Table 3). However, both the screening step assessment ETR and the first-tier assessment ETR calculated for flupyradifurone exceeded the trigger value for both acute and chronic oral exposure to bumblebees. This suggests that flupyradifurone does not meet the specific protection requirements for bumblebees, and a higher-tier risk assessment is necessary (Table 4).

## 4. Discussion

In light of the ban on various neonicotinoid insecticides, flupyradifurone has emerged as a potential alternative with significant prospects for the cultivation of protected crops. As a relatively new insecticide, it is crucial to conduct thorough toxicity tests and risk assessments of flupyradifurone on different pollinators to safeguard these species and encourage the sustainable use of this pesticide. Based on the toxicity and risk assessment results of this study, flupyradifurone does not meet the specific protection requirements for bumblebee pollinators. Furthermore, species-specific sensitivity was observed in the two bumblebee species tested. Therefore, flupyradifurone should be applied judiciously in agricultural practices to minimize harm to pollinators. Additionally, more species should be considered when conducting pesticide risk assessments.

Due to species-specific characteristics and test methods, the toxicity determination and risk assessment of the same pesticide can vary significantly across different organisms [41]. In our study, the two bumblebee species exhibited notable species-specific sensitivity to flupyradifurone in both acute and chronic exposure scenarios. Mundy-Heisz reported that the acute oral 48-h-LD_50_ value of flupyradifurone for *B. impatiens* was >1.7 μg/bee [42]. It is important to note that, due to the limitations in the pesticide concentration gradient used in Mundy-Heisz et al.’s study, the 48-h mortality rate in the highest dose treatment group (1.7 μg/bee) was 41.7%, still below the 50% mortality threshold [42]. Thus, while the LD_50_ value for bumblebees in their study was taken as the highest tested dose, the true LD_50_ should be higher than 1.7 μg/bee. Both Wang et al. [43] and Wang et al. [44] measured the acute oral toxicity of flupyradifurone to *B. terrestris*, but the 48-h LD_50_ values reported from these studies varied greatly, ranging from 3.508 μg/bee to 130.306 μg/bee. Notably, these studies used a honey bee toxicity measurement method, where different individuals were grouped in cages of 10 or 20. Unlike honey bees, bumblebees do not share food via trophallaxis [45], making it difficult to ensure that each individual consumes the same volume of sucrose solution when fed in groups. Additionally, hierarchy-related fighting often occurs among queenless bumblebee workers, which could contribute to mortality [46]. In contrast, our study employed the toxicity detection method for bumblebees developed by the OECD, which involves keeping individual bumblebees in separate breeding cages to ensure each bee is fed the same volume of test chemicals, providing more accurate responses to flupyradifurone. When conducting pesticide toxicity tests using bumblebees, it is crucial to standardize experimental apparatus and methods to ensure result accuracy. In this study, we report for the first time the chronic oral toxicity and NOAEC of flupyradifurone for these two bumblebee species. For wild pollinators such as bumblebees, which have smaller colonies, long-term exposure to pesticide residues during foraging is expected to be more detrimental than acute exposure. Chronic studies better represent the exposure of bees to pesticide residues in nectar or pollen, which is critical for risk assessment.

Most studies on flupyradifurone toxicity focus on the commercial bumblebee species *B. terrestris* and *B. impatiens*, with limited research on other species. In this study, we examined *B. lantschouensis*, a wild bumblebee species in China, and conducted toxicity and risk assessments, which is crucial for the protection and conservation of wild bumblebees. Our results showed that the toxicity of flupyradifurone to *B. lantschouensis* was higher than the toxicity to *B. terrestris*. Based on these findings, we propose three hypotheses. (1) *B. terrestris* may exhibit greater resistance to flupyradifurone than *B. lantschouensis,* potentially due to differences in detoxification and immune genes across species [47]. Bumblebee species face distinct evolutionary pressures, leading to variations in detoxification and immune gene sequences. These genetic differences might result in divergent responses to pesticide exposure under similar conditions. A comparative study of the detoxification and immune gene sequences between *B. terrestris* and *B. lantschouensis* is needed to determine whether these variations contribute to differences in insecticide sensitivity. (2) The differing responses of *B. terrestris* and *B. lantschouensis* to flupyradifurone may also be linked to the degree of domestication of each species. In this study, *B. terrestris* workers were artificially selected and domesticated, leading to improved success rates in artificial rearing, higher colony survival rates, and better pollination performance after multiple generations of positive selection [48]. This increased resilience enables *B. terrestris* to adapt more effectively to pollination tasks in controlled environments. In contrast, *B. lantschouensis* colonies used in this study were first-generation offspring from wild queens, with no artificial domestication or selection. As a result, these colonies demonstrated slightly lower success rates in artificial rearing, colony strength, and reproductive success under the same conditions compared to *B. terrestris* colonies [49]. We hypothesize that the resilience of *B. lantschouensis* is lower than that of *B. terrestris*. (3) The differences in sensitivity between *B. terrestris* and *B. lantschouensis* to flupyradifurone may be related to variations in body size and weight. Interspecific differences in responses to pesticide exposure are often linked to morphological variations. Based on our unpublished data, workers of *B. terrestris* have a heavier fresh weight and larger body size than *B. lantschouensis*. It has been reported that larger bees typically exhibit higher LD_50_ values and are less sensitive to a range of insecticides [50,51,52], and bee body size can influence sensitivity to pesticides [53]. The observed differences in toxicity and risk between *B. terrestris* and *B. lantschouensis* suggest that *B. terrestris* alone may not be a sufficiently representative species for bumblebee pesticide risk assessments. Therefore, it is important to include additional bumblebee species in the safety evaluation of new pesticides.

In the risk assessment, the HQ method indicated that flupyradifurone posed a low risk to bumblebees, while the ETR method suggested that both the acute and chronic oral toxicity of flupyradifurone were unacceptable, warranting higher-tier testing. Due to the different values of LD_50_ and application rate adopted in the equation, Mundy-Heisz et al.’s study classified flupyradifurone as a moderate risk using the HQ method but also emphasized the need for further testing using the ETR method [42]. The HQ method considers the amount of active ingredient in a commercial product applied to a crop and the acute LD_50_ of that ingredient, providing an initial estimate of the risk to insect pollinators from pesticide application. The ETR method, on the other hand, factors in nectar and pollen consumption, as well as worst-case pesticide residue levels, and is designed to assess whether further testing is required for a given pesticide. The discrepancy in risk categorization between these two methods underscores the need for more detailed pesticide use protocols to ensure the safety of pollinators. Much work remains to refine the foundational equations for risk assessment, particularly for non-*Apis* bees.

## 5. Conclusions

This study addresses the gap in understanding the lethal effects and risk assessments of flupyradifurone on the bumblebee species *B. terrestris* and *B. lantschouensis*, providing fundamental experimental data for future research. The effects of pesticides on bee species depend not only on intrinsic sensitivity but also on species-specific life cycles and foraging behaviors. The observed species-specific responses to flupyradifurone suggest that neither the wild bumblebee *B. lantschouensis* nor the commercial bumblebee *B. terrestris* can be considered representative of other bumblebee species. Further research is needed to better understand the interspecific variations in pesticide sensitivity among bumblebee taxa, and more bumblebee species should be included in pesticide risk assessments.

## Figures and Tables

**Figure 1 insects-16-00455-f001:**
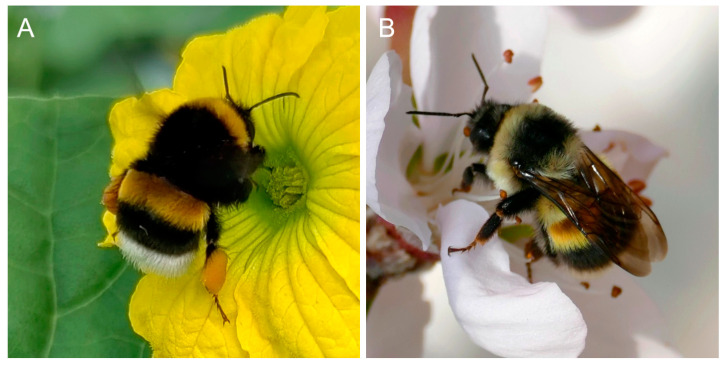
Bumblebee *Bombus terrestris* and *B. lantschouensis*. (**A**) Worker of *B. terrestris* visiting a melon flower; (**B**) worker of *B. lantschouensis* visiting a peach flower.

**Figure 2 insects-16-00455-f002:**
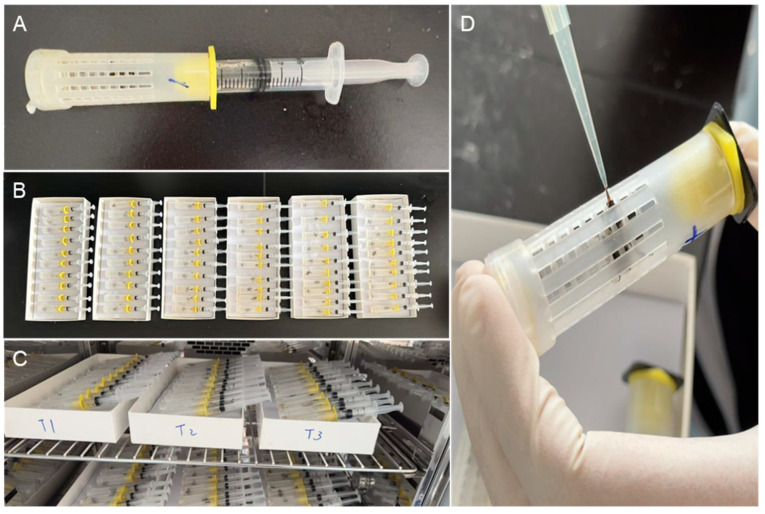
Experimental setup for the bumblebee oral toxicity test. (**A**) Single housing cage; (**B**) trial setup; (**C**) bumblebee rearing in the incubator; (**D**) feeding bumblebee with sucrose solution containing the test chemical.

**Figure 3 insects-16-00455-f003:**
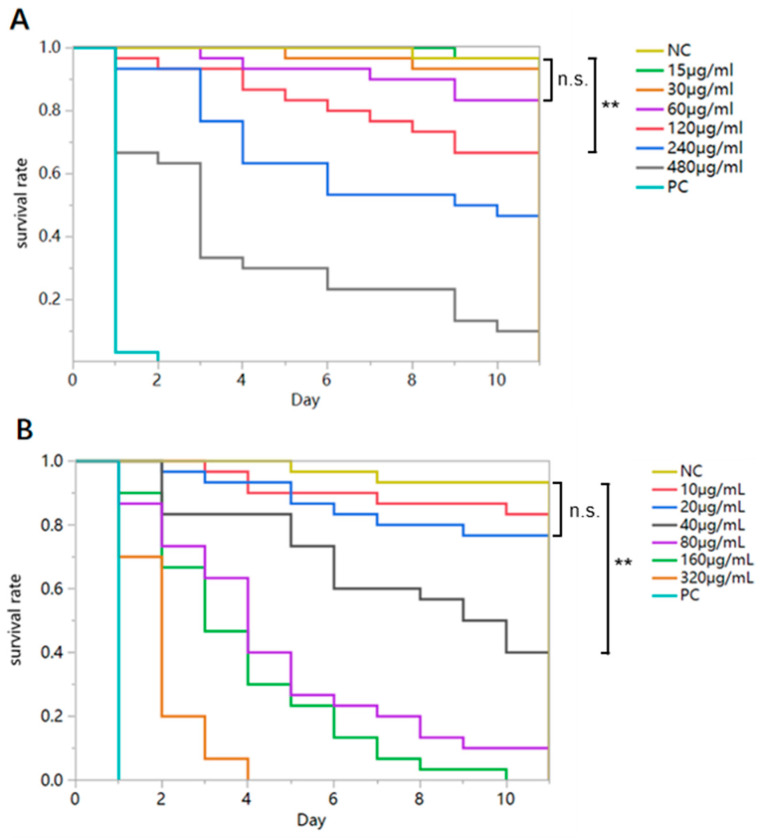
Survival of bumblebee workers exposed to different concentrations of flupyradifurone. (**A**) Survival rate of *B. terrestris* workers following long-term exposure to various concentrations of flupyradifurone (30 bees per treatment concentration); (**B**) Survival rate of *B. lantschouensis* workers following long-term exposure to different concentrations of flupyradifurone (30 bees per treatment concentration). ‘NC’ denotes non-contaminated diet; ‘PC’ represents 200 μg/mL dimethoate; ‘n.s.’ indicates no significant difference (*p* > 0.05). ‘**’ indicates a highly significant difference (*p* < 0.01). The data analysis corresponds to Appendix A.

**Table 1 insects-16-00455-t001:** Acute oral toxicity of flupyradifurone to workers of the two bumblebee species.

Species	*n*	R^2^	LC_50_ (95%CI) ^a^ (μg/mL)	LD_50_ (95%CI) ^b^ (μg/bee)	Toxic Regression Equations
*Bombus terrestris*	210	0.89	1400 (1100~1700)	28 (22~35)	y = 3.72x − 6.68
*Bombus lantschouensis*	210	0.96	250 (180~330)	5.1 (3.7~7.0)	y = 1.71x + 0.88

Note: ‘a’ indicates the median lethal concentration (LC_50_) of bumblebee deaths, with 95% confidence intervals (CI) in brackets. ‘b’ indicates the median lethal dose (LD_50_) of bumblebee deaths, with 95% confidence intervals (CI) in brackets.

**Table 2 insects-16-00455-t002:** Chronic oral toxicity of flupyradifurone to workers of the two bumblebee species.

Species	*n*	R^2^	LDD_50_ (95%CI) ^a^ (μg/bee/day)	Toxic Regression Equations
*Bombus terrestris*	240	0.96	3.3 (2.2~4.8)	y = 1.98x + 0.62
*Bombus lantschouensis*	240	0.96	0.7 (0.5~0.8)	y = 2.84x + 0.69

Note: ‘a’ indicates the median lethal dietary dose (LDD_50_) of bumblebee deaths, with 95% confidence intervals (CI) in brackets.

**Table 3 insects-16-00455-t003:** Risk classification of flupyradifurone to workers of the two bumblebee species based on the Hazard Quotient (HQ) equation.

Species	AR (g/ha)	LD_50_ (μg/bee)	HQ Value	Risk Level
*Bombus terrestris*	102	28	4	Low
*Bombus lantschouensis*		5.1	20	Low

Note: AR: Recommended application rate, obtained from the maximum recommended field use rate of flupyradifurone within commercial products registered by the Ministry of Agriculture and Rural Affairs of the People’s Republic of China. LD_50_: The 72-h lethal dose for 50% of bumblebee deaths, obtained from acute oral toxicity testing. HQ: The ratio of AR-to-LD_50_. Risk level: Risk levels for the HQ are low (<50), moderate (50–2500), and high (>2500) [38].

**Table 4 insects-16-00455-t004:** Risk classification of flupyradifurone to workers of the two bumblebee species using the Exposure Toxicity Ratio (ETR) (screening step and first tier assessment) risk equations.

Assessment Type	Species	Application Method	Exposure Toxicity Ratio		Trigger Value
			Screening Step Assessment	First Tier Assessment	
Acute oral exposure	*Bombus terrestris*	Down-ward	0.0408	0.0408	0.036
		Side-ward	0.0485	0.0485	
	*Bombus lantschouensis*	Down-ward	0.224	0.224	
		Side-ward	0.266	0.266	
Chronic oral exposure	*Bombus terrestris*	Down-ward	0.346	0.249	0.0048
		Side-ward	0.411	0.296	
	*Bombus lantschouensis*	Down-ward	1.632	1.175	
		Side-ward	1.938	1.395	

## Data Availability

The original contributions presented in this study are included in the article/Appendix A. Further inquiries can be directed to the corresponding authors.

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
