# Peer review of "Flupyradifurone Exhibits Greater Toxicity to the Asian Bumblebee Bombus lantschouensis Compared to the European Bumblebee Bombus terrestris"

_insects, 2025, doi:10.3390/insects16050455_

Round 1

Reviewer 1 Report

Comments and Suggestions for Authors

The paper investigates the acute and chronic toxicity of Flupyradifurone on two species of bumblebees, Bombus terrestris and Bombus lantschouensis.

I have only one objection. In the section „Materials and Methods“ is missing text regarding Statistical Analysis. What statistical tests did you use? In which program data analysis was performed? What level of significance did you use? …… please describe in detail. Adequate statistical tests are mentioned in the text, but it should also be stated in the materials and methods section.

Author Response

Responses to Reviewer 1

Comments 1: I have only one objection. In the section “Materials and Methods” is missing text regarding Statistical Analysis. What statistical tests did you use? In which program data analysis was performed? What level of significance did you use? …… please describe in detail. Adequate statistical tests are mentioned in the text, but it should also be stated in the materials and methods section.

Response 1: Thank you for pointing this out and we agree with this comment. Therefore, we have added the ‘Statistical Analysis’ in the ‘Materials and Methods’ section. Please find the details in the revised manuscript (Line 200-205).

Reviewer 2 Report

Comments and Suggestions for Authors

Dear Authors,

I have no substantive revisions to your manuscript. Your study is based on the analysis of a large number of data that have been comprehensively analyzed.

However, I would recommend a number of corrections.

1) Table 1, Table 2, Table 3. In the notes I would recommend to give more clearly what means the abbreviations LC, LD and others, which are in the tables. Of course, it is obvious, but, nevertheless, the generally accepted rules require such explanation in the tables.  

2) In the introduction, I would recommend to give the brief mention in which regions Bombus lantschouensis is known, not just Asia in general. For example, you could insert this information in a sentence on lines 92 and 93. You may include the regions in parentheses. Of course, this information is easy for the reader to find in relevant papers, but it is better to have this information immediately in your article.

Author Response

Responses to Reviewer 2

Dear Authors,

I have no substantive revisions to your manuscript. Your study is based on the analysis of a large number of data that have been comprehensively analyzed.

However, I would recommend a number of corrections.

Comments 1: Table 1, Table 2, Table 3. In the notes I would recommend to give more clearly what means the abbreviations LC, LD and others, which are in the tables. Of course, it is obvious, but, nevertheless, the generally accepted rules require such explanation in the tables.

Response 1: Thank you for pointing this out and we agree with this comment. Therefore, we have added the details in the revised manuscript (Line 166, 188, 209, 214-216, 225-226, 262-263).

Comments 2: In the introduction, I would recommend to give the brief mention in which regions Bombus lantschouensis is known, not just Asia in general. For example, you could insert this information in a sentence on lines 92 and 93. You may include the regions in parentheses. Of course, this information is easy for the reader to find in relevant papers, but it is better to have this information immediately in your article.

Response 2: Thank you for pointing this out and we agree with this comment. Therefore, we have added the distribution information of B. lantschouensis in the revised manuscript (Line 93-94).

Reviewer 3 Report

Comments and Suggestions for Authors

Summary: This paper describes research conducted examining the acute and chronic lethality of flupyradifurone on two bumble bee species. This reviewer found this paper to be clear and well written with figures and tables that effectively encapsulated the major findings. The authors' findings should serve as an important contribution to the growing body of literature examining pesticide effects on pollinators. Below are minor suggestions for improvement:

- L100: This is minor but the abbreviation "ddH2O" refers to double distilled water not deioned water (diH2O)
- L107: Change both mentions of the word "was" to "were" since they are describing plural rather than singular subjects
- For Table 1, this is minor but if the authors could arrange the table so that the 95% confidence interval in parentheses is placed next to the LC50 value (making a single row for each bee species similar to Table 2) that would help with clarity. This reviewer was initially confused by the current set up 
- For Table 2, similarly to Table 1, consider adjusting the width of the table so that the column "LDD50 (95%CI)a (ug/bee/day) is arranged in a single row.
- The Discussion is thorough and well written.

Author Response

Responses to Reviewer 3

Summary: This paper describes research conducted examining the acute and chronic lethality of flupyradifurone on two bumble bee species. This reviewer found this paper to be clear and well written with figures and tables that effectively encapsulated the major findings. The authors' findings should serve as an important contribution to the growing body of literature examining pesticide effects on pollinators. Below are minor suggestions for improvement:

Comments 1: L100: This is minor but the abbreviation "ddH2O" refers to double distilled water not deioned water (diH2O)

Response 1: Thank you for pointing this out and we agree with this comment. We have corrected this error in the revised manuscript. (Line 103).

Comments 2: L107: Change both mentions of the word "was" to "were" since they are describing plural rather than singular subjects

Response 2: Thank you for pointing this out and we agree with this comment. We have corrected this error in the revised manuscript. (Line 109).

Comments 3: For Table 1, this is minor but if the authors could arrange the table so that the 95% confidence interval in parentheses is placed next to the LC50 value (making a single row for each bee species similar to Table 2) that would help with clarity. This reviewer was initially confused by the current set up 

Response 3: Thank you for pointing this out and we agree with this comment. We have reformatted Table 1 as suggested (with LC50 values and their 95% confidence intervals presented in single rows for each bee species, consistent with Table 2's layout). (Line 213).

Comments 4: For Table 2, similarly to Table 1, consider adjusting the width of the table so that the column "LDD50 (95%CI)a (ug/bee/day) is arranged in a single row.

Response 4: Thank you for pointing this out and we agree with this comment. We have reformatted Table 2 as suggested (Line 224).

Comments 5: The Discussion is thorough and well written.

Response 5: Thank you.